# Topological Retrieval-Augmented Generation via Intersecting Evidence Paths

## Abstract

Retrieval-Augmented Generation (RAG) struggles with complex queries. While multi-query rewriting enhances recall by capturing diverse semantic dimensions, existing methods falter by consolidating retrieved documents into a flat list for reranking. This discards the crucial structural information from the rewriting process and fails to prioritize documents that bridge different query aspects. To address this issue, we propose HPT-TRACE, a framework that centers on a novel topology-aware reranking mechanism. This framework functions within a topological space defined by our Hierarchical Partition Tree (HPT), which is construction-efficient and does not rely on Large Language Models (LLMs). Our innovative Topological Reranking via Ancestor Convergence Evaluation (TRACE) algorithm operates within this HPT-defined space. Rather than scoring documents in isolation, TRACE considers each document's lineage in the tree as an evidence path. It then reranks candidates by assessing the intersection length of evidence paths originating from different semantic dimensions of the user's query. A document is deemed essential for synthesizing a comprehensive answer if its path contributes to an intersection of substantial length. By explicitly modeling the relationships between intersecting evidence paths, HPT-TRACE provides a framework that is both highly effective and computationally efficient, excelling at identifying the most salient and holistic information to significantly enhance retrieval for complex queries.

## 1 Introduction

Retrieval-Augmented Generation (RAG) enhances Large Language Models (LLMs) by grounding them in external documents, improving factual accuracy and mitigating hallucinations (Huang et al., 2025; Zamani & Bendersky, 2024; Lewis et al., 2020). However, RAG systems face a significant hurdle with long-context reasoning and complex queries, which demand deep contextual understanding and the synthesis of information scattered across lengthy texts. While advances like dense retrievers (Qu et al., 2020; Nian et al., 2025), dual-encoder models (Guu et al., 2020; Liu et al.), and memory-augmented networks (Borgeaud et al., 2022; Santoro et al., 2016) have significantly improved semantic retrieval, they often fall short of providing the structured, synthesized evidence needed to construct a comprehensive answer. Consequently, a core challenge remains in moving beyond simple relevance retrieval to explicitly identifying and synthesizing the disparate but interconnected evidence required for complex reasoning.

While Multi-Query Rewriting (Li et al., 2024; Kostric & Balog, 2024) is an effective strategy for improving recall on complex queries by probing a knowledge base from multiple perspectives to uncover a rich pool of candidate evidence, its potential is critically undermined by a flawed post-retrieval process. Standard pipelines merge the retrieved documents into a single, unstructured list and apply a reranker that scores relevance only against the original query. This merge-and-rerank approach discards the crucial structure of the multi-query process, ignoring which query retrieved which document. Consequently, it cannot identify documents that form conceptual bridges between query aspects, often resulting in a redundant context that lacks the synthesized information needed for a cohesive answer.

To address this limitation, we propose HPT-TRACE, a framework that fundamentally redesigns reranking for multi-query RAG. Rather than discarding the multi-query structure, we leverage it as a

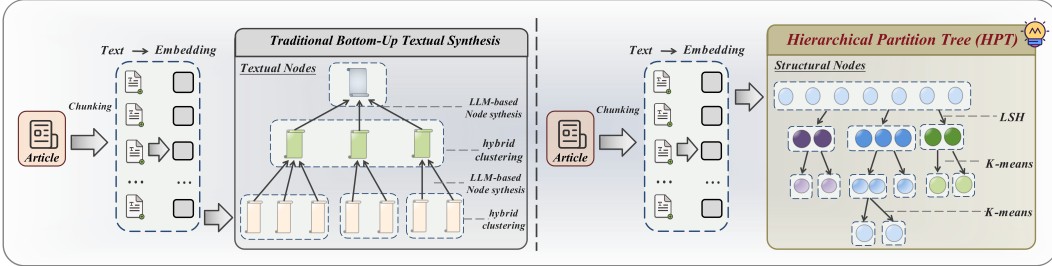

Figure 1: Contrasting paradigms for hierarchical structure construction over document embeddings. **Left**: Traditional *bottom-up* approaches recursively cluster nodes and use a LLM to synthesize new textual summary nodes, incurring significant computational costs. **Right**: Our HPT method applies an efficient, *top-down* divisive partitioning directly within the vector space. This creates a semantic hierarchy of non-textual, structural nodes without any LLM intervention during construction.

critical signal for identifying synthesis-oriented information. Our approach requires a shared topological space in which to analyze the relationships between documents retrieved from different query perspectives. To construct this space, we introduce the Hierarchical Partition Tree (HPT)(Fig. 1, right), a novel structure built via an efficient, *top-down* divisive clustering algorithm. Unlike conventional methods that rely on costly, LLM-based *bottom-up* synthesis (Fig. 1, left), HPT's LLM-free construction provides the necessary semantic hierarchy at a fraction of the computational cost, making our topology-aware framework practical and scalable.

The HPT provides a unified semantic hierarchy that serves as a shared space for all retrieved documents. Operating within this topological framework, our Topological Reranking via Ancestor Convergence Evaluation (TRACE) algorithm analyzes interrelations across the different evidence sets. Each document is represented as an evidence path. This path is defined as the unique sequence of nodes from the root to its leaf in the tree. Our key insight is that the most critical documents are those whose evidence paths deeply intersect with paths originating from multiple, distinct subqueries. While the intersection length between any two paths is measured by the depth of their lowest common ancestor (LCA) (Bender et al., 2005), our TRACE algorithm leverages this principle more broadly. For each candidate document, TRACE quantifies its conceptual alignment with every other evidence set by calculating convergence depth, defined as the depth of its strongest connection (i.e., deepest LCA) to any document within that set. The final ranking score is then aggregated from these individual convergence scores, prioritizing candidates that demonstrate consistent and substantial overlap across the diverse facets of the user's query. This holistic approach allows HPT-TRACE to construct a context that is not only relevant, but also synthesized cohesively to capture the multifaceted intent of the user. The effectiveness of this topology-aware approach is validated through extensive experiments on several challenging, reasoning-intensive benchmarks. Our main contributions are summarized as follows:

- We propose HPT, an efficient, LLM-free method for constructing a topological space designed specifically to enable the analysis of evidence paths.

- We propose TRACE, a novel topology-aware reranking algorithm that operates on the HPT, prioritizing a direct measurement of evidence path convergence over standard semantic similarity to identify documents crucial for building a comprehensive answer.

- We demonstrate the effectiveness of the complete HPT-TRACE framework, which consistently outperforms both leading retrieval frameworks and advanced reranking models on complex reasoning benchmarks. This performance validates its computationally efficient, path-intersection reranking as a novel approach that captures crucial inter-document relationships inherently missed by conventional methods that score documents in isolation.

## 2 RELATED WORK

**Retrieval-Augmented Generation Approaches.** RAG models enhance LLMs in open-domain QA and text generation by retrieving external knowledge and integrating it into the generation process (Huang & Huang, 2024; Asai et al., 2024; Cheng et al., 2025), thereby mitigating limitations

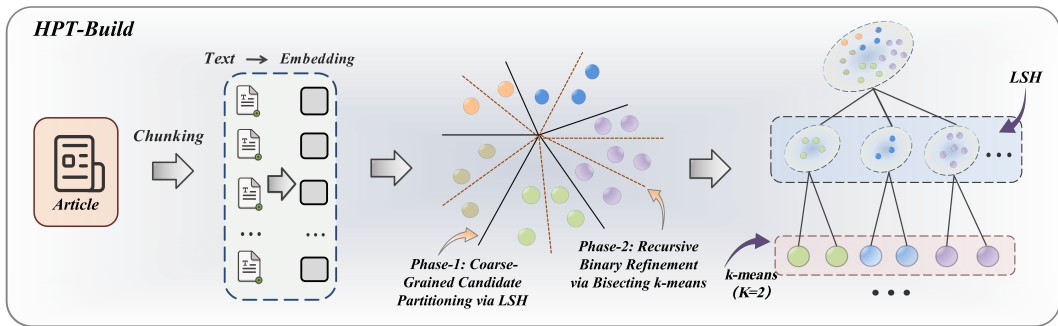

Figure 2: The pipeline of HPT construction. A *top-down*, two-phase hybrid strategy first uses Locality-Sensitive Hashing (LSH) for coarse-grained partitioning, followed by recursive Bisecting K-means to create a fine-grained binary hierarchy.

in domain-specific knowledge (Wu et al., 2024; Jiang et al., 2024; Guo et al., 2024). The initial RAG framework by Lewis et al. (2020) combined a retriever and generator, substantially improving QA accuracy. Subsequent work extended this paradigm: GraphRAG (Edge et al., 2024) models cross-document relations via graph structures to support multi-hop reasoning; PathRAG (Chen et al., 2025) performs path-level retrieval to improve contextual consistency. RAPTOR (Sarthi et al., 2024) employs a hierarchical tree-based mechanism for coarse-to-fine retrieval, improving both efficiency and precision in large-scale knowledge settings (Fatehkia et al., 2024).

**Optimization of Retrieval Methods.** Beyond framework-level improvements, another research direction focuses on optimizing the retrieval module. Early systems rely on lexical matching methods such as TF-IDF (Yun-tao et al., 2005) and BM25 (Robertson et al., 2009), with BM25 still widely used as an efficient and interpretable baseline. More recent work has shifted to dense retrieval, where neural encoders map queries and documents into continuous embeddings for semantic matching. Representative methods include DPR (Karpukhin et al., 2020), ANCE (Xiong et al., 2020), and ColBERT (Khattab & Zaharia, 2020), which significantly outperform lexical approaches. Hybrid strategies further combine sparse precision with dense generalization, achieving a better balance between efficiency and accuracy in large-scale RAG systems (Lin et al., 2021).

**Advanced Reranking Paradigms.** Advanced reranking is shifting from pointwise scoring to context-aware evaluation. Current trends employ LLMs for listwise reranking of candidate sets (Pradeep et al., 2023; Sun et al., 2023) or model document graphs to rerank based on semantic connectivity (Edge et al., 2024). These powerful paradigms, however, rely on intensive semantic computation at inference time. This work explores a distinct, topology-aware alternative by leveraging a pre-computed structural signal. Our proposed algorithm uses evidence path convergence to efficiently identify documents critical for synthesizing information, an approach orthogonal to content-focused scoring.

## 3 METHODOLOGY

Our HPT-TRACE framework operates as a cohesive, three-stage pipeline. The first stage is the offline construction of the HPT, which structures the corpus into a topological space. The second stage, Establishing Candidate Evidence Paths, maps the user's query onto this space to identify multiple potential lines of reasoning. The final stage is our novel TRACE algorithm, which analyzes the convergence of these candidate paths to perform a topology-aware reranking. We detail each of these integral stages below. A comprehensive summary of the notation used throughout this section is provided in Appendix I for reference.

### 3.1 HPT CONSTRUCTION

The HPT organizes the knowledge corpus into a directed tree, formally denoted as $\mathcal{T} = (\mathcal{V}, \mathcal{E})$. This structure enforces a hierarchical organization of data, progressing from abstract concepts at the root to fine-grained details in the lower layers. We distinguish between two types of nodes: leaf nodes,

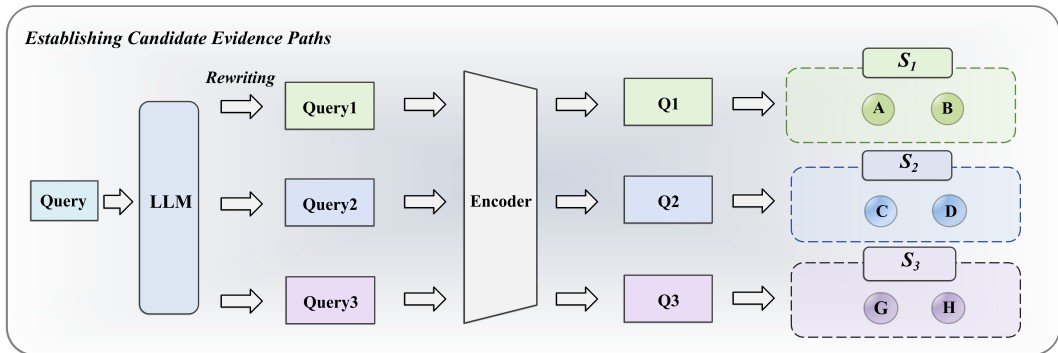

Figure 3: Mapping multi-query retrieval to evidence paths. Sub-queries $(Q_1, Q_2, Q_3)$ retrieve distinct evidence sets $(S_1, S_2, S_3)$. In the HPT's topological space, each document is represented as a unique root-to-leaf evidence path, setting the stage for TRACE to analyze path convergence.

which correspond to document chunks from the embedded document chunks $\mathcal{D} = \{d_1, ..., d_N\}$, are represented as $v_{\text{leaf}} = (id, p_v, \mathbf{w}_v, i_{\text{orig}})$, where $\mathbf{w}_v \in \mathbb{R}^d$ is its embedding vector anchoring the node in a high-dimensional semantic space, and $i_{\text{orig}}$ is its original corpus index. Internal nodes are purely structural elements represented as $v_{\text{internal}} = (id, p_v, \mathcal{C}_v, d_v)$, where $p_v$, $\mathcal{C}_v$, and $d_v$ are its parent pointer, child set, and depth, respectively. The root node of the tree is defined to have a depth of 0. The parent pointer and child set define the tree's topology, while the depth reflects the node's level of conceptual abstraction.

A fundamental design choice that distinguishes the HPT lies in its tree construction philosophy. Existing hierarchical methods, such as the RAPTOR framework, typically follow a *bottom-up abstraction* paradigm: they recursively cluster nodes and then employ a LLM to generate a textual summary for the new parent node. While powerful, this approach is computationally expensive and heavily reliant on LLMs. In a deliberate departure, HPT is constructed upon a *top-down refinement* paradigm. Instead of summarizing, our goal is to recursively partition the semantic space into smaller, more cohesive clusters. This process does not create new, abstract information; it refines and specifies existing semantic regions. The critical advantage of this approach is that it allows us to create a meaningful hierarchy for retrieval without resorting to LLM-based summarization or geometric synthesis for internal nodes. The hierarchy's value is derived from the topological paths it creates, which are directly exploited by our reranking algorithm.

This refinement philosophy dictates our choice of a hybrid clustering strategy. Fig. 2 provides a visual depiction of this pipeline. To efficiently construct the hierarchy from a massive set of $N$ embeddings of dimensionality $d$ while avoiding prohibitive computational complexity, our method combines a broad, coarse-grained partitioning phase using LSH with a fine-grained recursive subdivision using Bisecting K-Means. The complete algorithm, including recursive termination conditions, is formally presented in Appendix B (Algorithms 1 and 2). The resulting $O(N \cdot d \cdot \log N)$ time complexity ensures scalability, as formally derived in Appendix C.1. The specific implementation details, including our document chunking strategy and the configuration of our two-phase clustering pipeline, are provided in Appendix F.3.

## 3.2 ESTABLISHING CANDIDATE EVIDENCE PATHS

With the HPT established, the second stage of our framework, Establishing Candidate Evidence Paths, decomposes the user's information need into multiple lines of reasoning to retrieve a comprehensive set of candidate documents. As illustrated in Fig. 3, this process begins by augmenting the original user query $Q_1$ with a set of $M - 1$ diverse sub-queries generated by an LLM, resulting in a complete query set $\{Q_1, \ldots, Q_M\}$. Each query is designed to probe a distinct facet of the user's intent. These textual queries are then transformed into embedding vectors via an encoder. Finally, we execute a separate top-$k$ vector search for each query embedding against the document corpus, yielding a collection of distinct evidence sets, $\{\mathcal{S}_1, \ldots, \mathcal{S}_M\}$.

Crucially, instead of treating each retrieved document as an isolated item, we reconceptualize it topologically. We define a document's evidence path as the unique sequence of nodes from the root of the tree down to its corresponding leaf. This path represents the hierarchical chain of conceptual partitioning decisions that culminates in that single piece of evidence. For instance, as depicted in Fig. 4, the evidence path for document A is the unique sequence of nodes $\mathcal{R} \rightarrow K \rightarrow L \rightarrow A$, where $\mathcal{R}$ denotes the root of the tree. The union of all retrieved documents forms the candidate pool, *i.e.*, $\mathcal{S}_{\text{cand}} = \bigcup_{i=1}^{M} \mathcal{S}_i$, which is now understood not as a flat list, but as a collection of intersecting evidence paths. The core hypothesis of our framework is that a candidate's relevance to the user's original, multifaceted intent is strongly indicated by the extent to which its evidence path converges with paths originating from different sub-queries. Section 3.3 details our proposed TRACE algorithm, which formalizes a scoring mechanism to quantify this convergence.

## 3.3 TRACE: Topological Reranking via Convergence Depth

Given the evidence paths for all documents in the candidate pool $\mathcal{S}_{\text{cand}} = \bigcup_{i=1}^{M} \mathcal{S}_i$, the TRACE algorithm reranks these candidates by quantifying their conceptual convergence. The fundamental principle of our method is based on a core topological insight: the intersection of any two evidence paths is the shared path segment that extends from the root $\mathcal{R}$ to their LCA. Consequently, the length of this shared path, which we term the intersection length, is precisely measured by the depth of their LCA. A deeper LCA indicates a more specific and substantial shared conceptual context between the two documents. For instance, as illustrated in Fig. 4, the evidence paths for documents $\mathcal{A}$ and $\mathcal{C}$ intersect along the path $\mathcal{R} \rightarrow \mathcal{K}$. Their LCA is node $\mathcal{K}$, and the Intersection Length is accordingly given by depth(LCA($\mathcal{A}, \mathcal{C}$)) = depth($\mathcal{K}$) = 1.

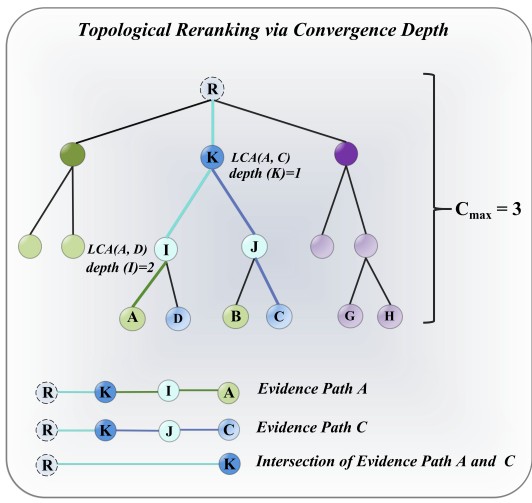

Figure 4: The TRACE scoring mechanism. The figure illustrates how a candidate's score is derived from its convergence depth with other evidence sets. This metric is fundamentally based on the depth of the LCA between evidence paths, which serves to quantify their conceptual overlap.

While pairwise overlap provides a useful signal, evaluating a candidate's relevance to a multifaceted query requires quantifying its alignment with an entire query facet, as represented by a full evidence set. To this end, we formalize the measure of convergence depth, $c_{l,i}$, which is the length of the longest shared evidence path between a candidate document $l$ and any document within a given evidence set $\mathcal{S}_i$:

$$c_{l,i} = \max_{l' \in \mathcal{S}_i} \left( \text{depth}(\text{LCA}(l, l')) \right) \tag{1}$$

A high $c_{l,i}$ value directly corresponds to a long shared evidence path, signifying a deep and substantial conceptual overlap between the candidate $l$ and the semantic facet represented by $\mathcal{S}_i$.

With this metric, we can design a final scoring function that rewards candidates demonstrating deep and consistent convergence with multiple query facets. Our design philosophy prioritizes two key principles. First, we apply a quadratic reward to disproportionately amplify the scores of documents that serve as strong conceptual bridges. Second, and crucially, a document's score is derived from its ability to connect with *other* query facets. To produce a standardized score that is comparable across different queries, we then average this value across the query facets. A candidate document $l \in \mathcal{S}_j$ may also appear in other sets; to maintain a consistent scoring logic, its final TRACE score is defined as the average of its squared normalized convergence depths across all $M$ query facets. Let $C_{\max} = \max_{l,i} c_{l,i}$ be the observed global maximum convergence depth. The score for any candidate $l$ is:

$$\text{Score}(l) = \frac{1}{M} \sum_{i=1}^{M} \left( \frac{c_{l,i}}{C_{\max}} \right)^2 \tag{2}$$

When evidence paths from different sub-queries are topologically isolated, TRACE's signal weakens, and the system gracefully degrades to its semantic similarity tie-breaker. A detailed analysis of this limitation alongside the framework's successes is provided in Appendix G.

For a concrete illustration, we calculate the TRACE score for candidate $\mathcal{A}$ based on the HPT shown in Fig. 4. Assume our process yields $M = 3$ query sets: $\mathcal{S}_1 = \{\mathcal{A}, \mathcal{B}\}$, $\mathcal{S}_2 = \{\mathcal{C}, \mathcal{D}\}$, and $\mathcal{S}_3 = \{\mathcal{G}, \mathcal{H}\}$. The necessary convergence depths for candidate $\mathcal{A}$ are:

$$c_{\mathcal{A},1} = \max(\text{depth}(\text{LCA}(\mathcal{A}, \mathcal{A})), \text{depth}(\text{LCA}(\mathcal{A}, \mathcal{B}))) = \max(3, 2) = 3 \tag{3}$$

$$c_{\mathcal{A},2} = \max(\text{depth}(\text{LCA}(\mathcal{A}, \mathcal{C})), \text{depth}(\text{LCA}(\mathcal{A}, \mathcal{D}))) = \max(1, 2) = 2 \tag{4}$$

$$c_{\mathcal{A},3} = \max(\text{depth}(\text{LCA}(\mathcal{A}, \mathcal{G})), \text{depth}(\text{LCA}(\mathcal{A}, \mathcal{H}))) = \max(0, 0) = 0 \tag{5}$$

With a global maximum convergence depth $C_{\max} = \text{depth}(\text{LCA}(\mathcal{D}, \mathcal{B})) = 3$, the final TRACE score for $\mathcal{A}$ is:

$$\text{Score}(\mathcal{A}) = \frac{1}{3}\left(\left(\frac{3}{3}\right)^2 + \left(\frac{2}{3}\right)^2 + \left(\frac{0}{3}\right)^2\right) \approx 0.48 \tag{6}$$

This score holistically quantifies $\mathcal{A}$'s relevance, driven by its self-identity in $\mathcal{S}_1$ and strong conceptual link to $\mathcal{S}_2$. In contrast, an isolated candidate like $\mathcal{G}$ from $\mathcal{S}_3$ would have convergence depths $c_{\mathcal{G},1} = 0$, $c_{\mathcal{G},2} = 0$, and $c_{\mathcal{G},3} = 3$, resulting in a lower score of $\text{Score}(\mathcal{G}) \approx 0.33$. This disparity highlights how TRACE effectively prioritizes documents that bridge multiple, distinct query facets.

While a naive implementation of Eq 1 would be computationally prohibitive, we employ an efficient approach using Path Encoding combined with Prefix Trees (Tries) (Aoe et al., 1992). The path from the root to each leaf is encoded as a sequence, and finding the LCA depth becomes equivalent to finding the longest common prefix (LCP) length between two paths. This transforms the graph traversal problem into a highly efficient string matching operation. This ensures TRACE is a lightweight online operation with negligible inference overhead, as analyzed in Appendix C.2. The full implementation is detailed in Algorithm 3.

Finally, all documents in the candidate pool $\mathcal{S}_{\text{cand}}$ are sorted in descending order based on their final TRACE score. In our implementation, any ties are broken using the raw semantic similarity score of the document to the original query, $Q_1$. This serves as a simple and effective heuristic for ordering topologically equivalent candidates, though we note this is a modular step where any standard reranking model could be integrated.

## 4 EXPERIMENTS

### 4.1 EXPERIMENTAL SETUP

**Benchmarks.** To comprehensively validate HPT-TRACE, we select a suite of benchmarks that probe distinct and challenging aspects of long-context question answering. These benchmarks include NarrativeQA (Kočiský et al., 2018), a reading comprehension benchmark designed to test a model's ability to understand long-form narratives from books and scripts; QASPER (Dasigi et al., 2021), a dataset focused on synthesizing information scattered across dense scientific documents; and QuALITY (Pang et al., 2021), for which we use its challenging HARD subset to evaluate deep, multi-hop reasoning. For our focused component analysis, we use HotpotQA (Yang et al., 2018), whose multi-hop question structure is the ideal environment for dissecting the efficacy of retrieval and reranking components.

**Evaluation Metrics.** Our evaluation metrics are chosen to align with the standard practices for each benchmark. For the abstractive, free-form answers in NarrativeQA, we report generation quality using ROUGE-L, which measures recall based on the longest common subsequence; BLEU-1 and BLEU-4 for n-gram precision; and METEOR, which considers synonymy and stemming. For QASPER and HotpotQA, we report the standard Answer F1 score, a robust metric for short-form QA that computes the token-level harmonic mean of precision and recall. For the multiple-choice format of QuALITY, we report Accuracy.

**Baselines.** We compare HPT-TRACE against a hierarchy of strong baseline models to situate its performance. Our primary state-of-the-art baseline for end-to-end comparison is RAPTOR (Sarthi et al., 2024), a retrieval method that recursively clusters and summarizes document chunks to build a

| Dataset | Model | ROUGE-L ↑ | BLEU-1 ↑ | BLEU-4 ↑ | METEOR ↑ |
|---|---|---|---|---|---|
| NarrativeQA | BiDAF (Kocisky et al., 2018) | 6.2 | 5.7 | 0.3 | 3.7 |
| | Rec. Summarizing + UQA (Wu et al., 2021) | 21.6 | 22.3 | 4.2 | 10.6 |
| | RAPTOR + UnifiedQA (Sarthi et al., 2024) | 30.8 | 23.5 | 6.4 | 19.1 |
| | **HPT-TRACE + UnifiedQA** | **32.5** | **24.1** | **6.8** | **19.9** |

| Dataset | Model | Answer F1 ↑ | |
|---|---|---|---|
| QASPER | LongT5 XL (Guo et al., 2022) | 53.1 | |
| | CoLT5 XL (Ainslie et al., 2023) | 53.9 | |
| | RAPTOR + GPT-4 (Sarthi et al., 2024) | 55.7 | |
| | **HPT-TRACE + GPT-4** | **56.8** | |

| Dataset | Model | Accuracy (Test) ↑ | Accuracy (Hard) ↑ |
|---|---|---|---|
| QuALITY | CoLISA (DeBERTaV3-L) (Dong et al., 2023a) | 62.3 | 54.7 |
| | RAPTOR + UnifiedQA (Sarthi et al., 2024) | 56.6 | - |
| | RAPTOR + GPT-4 (Sarthi et al., 2024) | 82.6 | 76.2 |
| | **HPT-TRACE + UnifiedQA** | **57.8** | **48.9** |
| | **HPT-TRACE + GPT-4** | **84.2** | **78.0** |

Table 1: End-to-end performance comparison against SOTA models on NarrativeQA, QASPER, and QuALITY benchmarks. HPT-TRACE consistently outperforms existing methods. The best results are shown in bold.

hierarchical index. For our detailed component analysis on HotpotQA, we compare against several paradigmatic retrieval and reranking strategies to precisely evaluate the contribution of TRACE. These include a foundational Standard RAG pipeline; Reciprocal Rank Fusion (RRF) (Cormack et al., 2009), an algorithmic baseline that fuses ranked lists from multiple queries; and a powerful Cross-Encoder Reranker (Pradeep et al., 2022), which serves as a strong semantic similarity benchmark. Additional baselines from the literature are included in our main results table for a comprehensive SOTA comparison.

**Implementation Details.** To ensure a fair and direct comparison, our entire experimental setup is designed to mirror the protocol established by Sarthi et al. (2024). All document chunks are embedded using the SBERT model sentence-transformers/all-mpnet-base-v2 (Reimers & Gurevych, 2019). We report results using two powerful reader models, UnifiedQA-3B and GPT-4, to demonstrate our framework's effectiveness across different generators. For HPT construction, our hybrid clustering uses LSH with 20 bands and 10 hashes per band, and a leaf node threshold of 30 chunks. During reranking, complex queries are augmented with four rewrites generated by GPT-4 via a one-shot prompt (see Appendix E), resulting in a total of $M = 5$ sub-queries (the original plus four rewrites). The initial candidate pool for each sub-query is set to $k = 15$. A complete list of all hyperparameters is provided in Appendix F.1 for full reproducibility.

## 4.2 MAIN RESULTS

Tab. 1 presents the end-to-end performance of HPT-TRACE against SOTA baselines, demonstrating consistently superior performance across three challenging long-context benchmarks. When paired with the open-source UnifiedQA reader, our method already shows a clear advantage on NarrativeQA, a benchmark that tests long-range plot understanding. It achieves a ROUGE-L score of 32.5, outperforming the strong RAPTOR baseline by 1.7 scores, which suggests our *top-down* partitioning creates a highly coherent context for reasoning. The core advantages of our architecture become particularly evident on benchmarks that necessitate synthesizing disparate evidence. On QASPER, which requires answering questions from dense academic papers, HPT-TRACE paired with GPT-4 achieves a new SOTA F1 score of 56.8. This advantage is further magnified on the QuALITY benchmark's HARD subset, designed for multi-hop reasoning, where our framework improves upon the powerful RAPTOR+GPT-4 baseline by a significant 1.8 scores. This consistent outperformance on reasoning-intensive benchmarks serves as direct validation of our core thesis.

To isolate the contribution of our TRACE algorithm, we compare it against a hierarchy of reranking strategies on the multi-hop HotpotQA benchmark (Tab. 2). A Direct Retrieval pipeline using a single query establishes a baseline of 68.5 F1. While simply expanding to multiple queries with a basic dense reranker (Base Retriever Reranking) offers a modest gain, more advanced algorithmic

| Reranking Paradigm | Key Components | Answer F1 ↑ |
|---|---|---|
| **HPT-TRACE** | **Multi-Query + Topology-Aware Reranking** | **77.6** |
| Cross-Encoder Reranker | Multi-Query + Semantic Cross-Encoder | 75.8 |
| Reciprocal Rank Fusion (RRF) | Multi-Query + Algorithmic Rank Fusion | 74.0 |
| Base Retriever Reranking | Multi-Query + Rerank with Base Retriever | 70.5 |
| Direct Retrieval | Single-Query + Base Retriever | 68.5 |

Table 2: SOTA comparisons on the HotpotQA dev set. Our topology-aware framework significantly outperforms strong semantic and algorithmic reranking baselines, demonstrating the value of modeling evidence path intersections.

| Model Variant | Description | Answer F1 ↑ | Time (s) ↓ |
|---|---|---|---|
| **HPT-TRACE (Full)** | **Our LSH + Bisecting K-Means** | **77.6** | **18.1** |
| w/o TRACE | Multi-Query + Naive Dense Reranking | 70.5 | 18.1 |
| w/o LSH (K-Means only) | Removes LSH pre-partitioning | 76.2 | 27.1 |

Table 3: Ablation and efficiency analysis of HPT construction components on the HotpotQA dev set. All variants use the TRACE reranking algorithm.

fusion (RRF) and powerful semantic reranking (Cross-Encoder) are required to push performance to 74.0 and 75.8, respectively. Crucially, by replacing semantic scoring with our topological analysis, HPT-TRACE achieves a final score of 77.6, outperforming the strong Cross-Encoder baseline by a significant 1.8 points. This result demonstrates that the topological signal of evidence path convergence provides a unique and powerful advantage for identifying synthesis-oriented documents that even advanced semantic models cannot capture.

### 4.3 ABLATION STUDIES AND ANALYSIS

To quantify the contributions of our core components, we now dissect the framework's architecture on the HotpotQA benchmark. Our analysis first validates the critical impact of the TRACE algorithm itself before examining the efficiency and robustness of the underlying HPT structure.

**The Critical Impact of TRACE Reranking.** The most striking result of our analysis emerges from a direct ablation of the TRACE component, as shown in Tab. 3. Removing TRACE and reverting to a standard dense reranker (*w/o TRACE*) causes the F1 score to plummet from 77.6 to 70.5. This sharp 7.1-point degradation isolates the substantial and indispensable value added by our topology-aware approach, confirming it is the primary driver of the framework's performance.

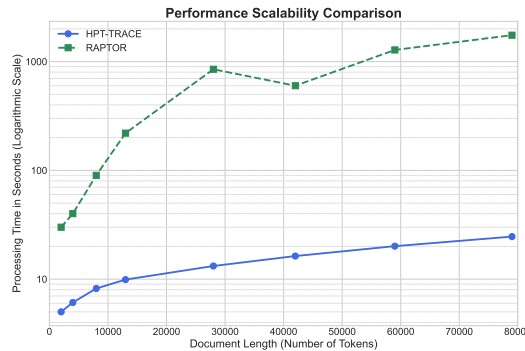

Figure 5: Scalability of HPT construction. We compare construction time (log-scale) of our LLM-free method HPT-TRACE against the LLM-based RAPTOR method.

**Efficiency and Scalability of LLM-Free HPT Construction.** A primary motivation for our HPT design is to achieve computational efficiency by avoiding LLMs during index construction. As shown in Tab. 3, our hybrid approach (LSH + K-Means) is not only 33% faster than using K-Means alone but also improves the final F1 score by 1.4, validating our design choices. More critically, this LLM-free design provides a fundamental scalability advantage over recent methods like RAPTOR. As illustrated in Fig. 5, the construction time of HPT-TRACE scales near-linearly with corpus size. In contrast, the LLM-based summarization in RAPTOR incurs rapidly escalating costs, making it

(a) Experimental results under different bands and hashes per band. The choice of (20, 10) achieves the optimal trade-off between performance and efficiency.

| Bands ($b$) | Hashes per Band ($k$) | Answer F1 ↑ | Construction Time (s) ↓ |
|---|---|---|---|
| 10 | 5 | 77.1 | 21.6 |
| 15 | 5 | 77.3 | 19.5 |
| **20** | **10** | **77.6** | **18.1** |
| 20 | 15 | 76.7 | 16.5 |
| 30 | 15 | 74.5 | 14.8 |
| 40 | 20 | 72.2 | 12.9 |

(b) Experimental results indicate that a leaf_threshold of 30 affords the optimal performance–efficiency trade-off.

| Leaf_Threshold | Answer F1 ↑ | Construction Time (s) ↓ |
|---|---|---|
| 10 | 76.5 | 57.3 |
| 20 | 77.3 | 35.5 |
| **30** | **77.6** | **18.1** |
| 50 | 76.8 | 15.2 |
| 75 | 75.3 | 13.9 |
| 150 | 72.9 | 11.5 |

Table 4: Hyperparameter analysis for HPT construction on the HotpotQA dev set. (a) The trade-off between Answer F1 and construction time for LSH parameters. (b) A similar trade-off analysis for the *leaf_threshold* in the recursive K-Means phase.

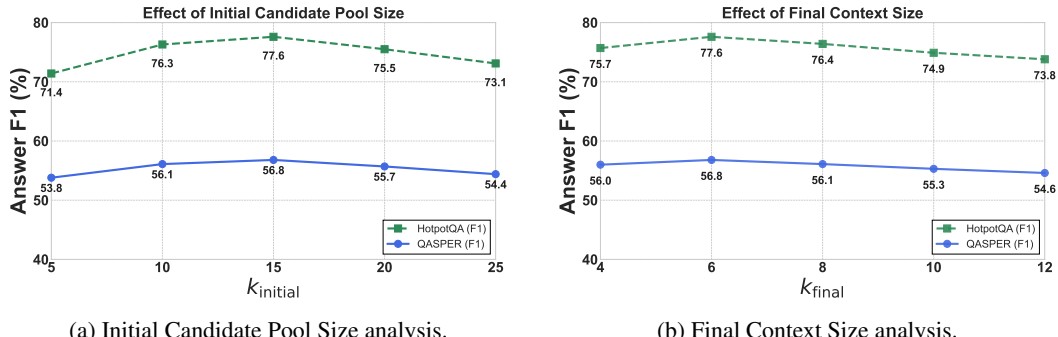

(a) Initial Candidate Pool Size analysis.   (b) Final Context Size analysis.

Figure 6: Analysis of retrieval and context sizes on HotpotQA and QASPER datasets. (a) The effect of the initial candidate pool size. (b) The effect of the final context size provided to the generator.

orders of magnitude slower and prohibitively expensive for large-scale, real-world RAG applications where indexing time and cost are critical constraints.

**Framework Robustness and Hyperparameter Sensitivity.** Finally, we investigate the sensitivity of HPT-TRACE to its key hyperparameters to validate its robustness. The analysis, conducted on the HotpotQA development set, shows that our framework is stable across a wide range of settings. For HPT construction, as shown in Tab. 4, our chosen LSH parameters (bands = 20, hashes per band = 10, leaf_threshold = 30) reside within a broad performance plateau, effectively balancing retrieval quality and indexing speed. The theoretical justification for our LSH parameter choice is detailed in Appendix F.2. Similarly, for TRACE retrieval, as shown in Fig. 6, performance remains high and stable around our default initial candidate pool size and final context size. This demonstrates that our framework's strong performance is not the result of brittle hyperparameter tuning but stems from its core architectural design. Further robustness tests presented in Appendix H, which evaluate the impact of alternative embedding models, confirm that the architectural benefits of HPT-TRACE are complementary to the quality of the underlying semantic space.

## 5 CONCLUSION

In this work, we present HPT-TRACE, a framework that shifts the reranking paradigm from isolated relevance scoring to holistic, topology-aware analysis. This is accomplished through two key innovations: the HPT, which provides an efficiently constructed semantic hierarchy via *top-down* divisive clustering without relying on LLMs, and the TRACE algorithm, which operates on this structure. By measuring the intersection of evidence paths using the depth of their LCA, TRACE provides a robust signal for synthesis-oriented reranking. Our evaluation demonstrates that this path-intersection criterion enables HPT-TRACE to consistently outperform advanced reranking models on reasoning-intensive benchmarks, while being significantly more efficient than *bottom-up*, summary-driven hierarchical methods.

## REPRODUCIBILITY STATEMENT

We are committed to the full reproducibility of our research. The core algorithms of our framework, HPT construction and the TRACE reranker, are detailed with pseudocode in Appendix B. To facilitate direct replication and further research, we will make our source code publicly available upon publication. Our experimental setup, including the benchmarks and evaluation metrics used, is described in Section 4. All hyperparameters necessary to reproduce our results, covering HPT construction, LSH parameters, and retrieval settings, are comprehensively listed in Appendix F.1. The structured prompt used for multi-query rewriting is provided verbatim in Appendix E. The datasets utilized in our experiments are all publicly available benchmarks: NarrativeQA, QASPER, QuALITY, and HotpotQA. Furthermore, our claims regarding the computational efficiency of our methods are supported by a formal complexity analysis, which is detailed in Appendix C. All experiments were conducted on a multi-GPU system with eight NVIDIA GeForce RTX 4060s, using a Python-based environment built upon core libraries such as PyTorch, Faiss, and key components from Hugging Face, including the Transformers and Datasets libraries.

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

## A  THE USE OF LARGE LANGUAGE MODELS (LLMS)

The authors utilized Google's Gemini 2.5 Pro to improve the grammar and clarity of the text. The scientific contributions and all analyses presented are the original work of the authors, who bear full responsibility for the paper's content.

## B  CORE ALGORITHMS

This section provides the detailed pseudocode for the primary components of the HPT-TRACE framework. The algorithms are designed to be both effective in their purpose and computationally efficient in their execution.

### B.1  HPT CONSTRUCTION PSEUDOCODE

The HPT is constructed via a top-down, LLM-free paradigm. This approach prioritizes scalability and avoids the significant computational costs associated with LLM-based summary generation. The process combines a coarse-grained global partitioning using Locality-Sensitive Hashing (LSH) with a fine-grained recursive subdivision using Bisecting K-Means. Algorithm 1 serves as the main entrypoint, orchestrating the two-phase construction, which in turn calls the recursive core function, BuildSubtree, presented in Algorithm 2.

---

**Algorithm 1** HPT Construction Entrypoint

**Input:** Corpus of embeddings $\mathcal{D}$, parameters $P$.
$\mathcal{T} \leftarrow \text{InitializeTree}()$
\# Create a structural root node (depth=0).
$\text{root} \leftarrow \text{CreateNode}(\text{parent=null,depth=0})$
$\mathcal{T}.\text{SetRoot}(\text{root})$
\# Phase 1: Coarse partition corpus with LSH.
$\{\mathcal{B}_i\} \leftarrow \text{ApplyLSH}(\mathcal{D}, P.lsh\_bands, P.lsh\_hashes\_per\_band)$
**for** each bucket of indices $\mathcal{B}_i$ in $\{\mathcal{B}_i\}$ **do**
  $\mathcal{D}_i \leftarrow \mathcal{D}[\mathcal{B}_i]$
  \# Create L1 node for each bucket.
  $n_i \leftarrow \text{CreateNode}(\text{parent=root}, \text{depth=1})$
  \# Phase 2: Recursively build subtree.
  $\text{BuildSubtree}(n_i, \mathcal{D}_i, P)$
**end for**

**Output:** The fully constructed HPT $\mathcal{T}$.

---

**Algorithm 2** Recursive Subtree Construction

**Input:** Parent node $n_\text{p}$, data points $\mathcal{P}_\text{part}$, params $P$.
\# Data point is a tuple (embedding, original_index).
**if** $|\mathcal{P}_\text{part}| \leq P.\text{leaf\_threshold}$ **then**
  **for** each point $(\mathbf{w}_j, i_j)$ in $\mathcal{P}_\text{part}$ **do**
    $\text{CreateLeafNode}(n_\text{p}, \mathbf{w}_j, i_j)$
  **end for**
  **return**
**end if**
\# Partition points into two clusters via K-Means.
$\mathcal{P}_0, \mathcal{P}_1 \leftarrow \text{PartitionWithKMeans}(\mathcal{P}_\text{part}, k = 2)$
**for** each cluster $\mathcal{P}_k$ in $(\mathcal{P}_0, \mathcal{P}_1)$ **do**
  **if** $|\mathcal{P}_k| = 0$ **then continue**
  **end if**
  $n_\text{c} \leftarrow \text{CreateNode}(n_\text{p}, n_\text{p}.\text{depth} + 1)$
  $\text{BuildSubtree}(n_\text{c}, \mathcal{P}_k, P)$
**end for**

---

### B.2  TRACE RERANKING WITH PATH ENCODING AND TRIES

A naive implementation of the TRACE scoring function, as defined in Equation 1, would be computationally prohibitive for online inference. It would require numerous pairwise computations of the LCA within nested loops, leading to a complexity that scales poorly with the size of the candidate pool. To overcome this bottleneck and ensure that reranking is a lightweight operation, we employ a two-part strategy based on *path encoding* and *prefix trees* (Tries).

First, during an offline preprocessing step that follows the HPT construction, we compute a unique path encoding for each leaf node in the tree. This encoding is simply the sequence of node identifiers from the root to that leaf, effectively translating the topological position of a document into a sequential data structure. Second, during online inference, we leverage this representation by building a separate Trie for each of the $M$ retrieved evidence sets. Each Trie stores the path encodings of all documents within its corresponding set. The critical insight is that the depth of the LCA between any two documents corresponds precisely to the length of the longest common prefix (LCP) of their path encodings. By querying a candidate document's path encoding against an evidence set's Trie, we can find the maximum LCP length—and thus the convergence depth—in time proportional to the path length, not the size of the evidence set. This transforms the problem from a complex

graph traversal task into a highly efficient string prefix matching operation. Algorithm 3 details this complete, optimized implementation.

---

**Algorithm 3** TRACE Reranking with Path Encoding and Tries

---

**Input:** HPT Tree $\mathcal{T}$ (with path encodings), Query Set $\{Q_1, \ldots, Q_M\}$, Parameters $P$
  # Step 1: Diverse Multi-Query Retrieval.
  $\mathcal{S}_{list} \leftarrow \text{RetrieveInitialSets}(\{Q_1, \ldots, Q_M\}, \mathcal{T}, P.k_{\text{initial}})$
  $S_{\text{cand}} \leftarrow \bigcup_{S \in \mathcal{S}_{list}} S$
  # Step 2: Pre-compute Prefix Trees (Tries) for each evidence set.
  Tries $\leftarrow$ new array of size $M$
  **for** $i \leftarrow 1$ to $M$ **do**
    $\text{encodings}_i \leftarrow \{l.\text{encoding for } l \in \mathcal{S}_{list}[i-1]\}$
    $\text{Tries}[i-1] \leftarrow \text{BuildTrie}(\text{encodings}_i)$
  **end for**
  # Step 3: Compute Raw Convergence Depths efficiently using Tries.
  RawDepths $\leftarrow$ new map from leaf to vector
  $C_{\max} \leftarrow 0$
  **for** each leaf $l$ in $S_{\text{cand}}$ **do**
    $\mathcal{V}_l \leftarrow$ new vector of size $M$
    **for** $i \leftarrow 1$ to $M$ **do**
      $c_{l,i} \leftarrow \text{GetMaxPrefixLength}(\text{Tries}[i-1], l.\text{encoding})$
      $\mathcal{V}_l[i-1] \leftarrow c_{l,i}$
      $C_{\max} \leftarrow \max(C_{\max}, c_{l,i})$
    **end for**
    $\text{RawDepths}[l] \leftarrow \mathcal{V}_l$
  **end for**
  # Step 4: Compute Final Scores.
  FinalScores $\leftarrow$ new map from leaf to score
  **if** $C_{\max} > 0$ **then**
    **for** each leaf $l$ in $S_{\text{cand}}$ **do**
      score $\leftarrow 0$
      **for** $i \leftarrow 1$ to $M$ **do**
        $\hat{c}_{l,i} \leftarrow \text{RawDepths}[l][i-1]/C_{\max}$
        score $\leftarrow$ score $+ (\hat{c}_{l,i})^2$
      **end for**
      $\text{FinalScores}[l] \leftarrow \text{score}/M$
    **end for**
  **else**
    **for** each leaf $l$ in $S_{\text{cand}}$ **do**
      $\text{FinalScores}[l] \leftarrow 0$
    **end for**
  **end if**
  # Step 5: Sort candidates by final score, with similarity as tie-breaker.
  $R \leftarrow \text{Sort}(S_{\text{cand}}, \text{key}=(\text{FinalScores}[\cdot], \text{Similarity}(\cdot, Q_1)), \text{order}=\text{desc})$
  **return** $R[0 : P.k_{\text{final}}]$
  **Output:** A reranked list of leaf nodes $R$

---

## C  COMPLEXITY AND EFFICIENCY ANALYSIS

A primary design consideration for the HPT-TRACE framework is computational efficiency, ensuring its applicability to massive, real-world corpora. This section provides a formal analysis of the time complexity for both the offline HPT construction phase and the lightweight online TRACE reranking algorithm.

### C.1  TIME COMPLEXITY OF HPT CONSTRUCTION

The construction of the HPT is an offline process designed for scalability. Its time complexity is determined by two sequential phases. Let $N$ be the total number of document chunks in the corpus and $d$ be the dimensionality of their corresponding embeddings.

The first phase, coarse-grained partitioning via Locality-Sensitive Hashing (LSH), is near-linear. The process involves generating hash signatures for each of the $N$ embeddings and grouping them into buckets. The cost of this operation is proportional to the number of embeddings, their dimension, and the number of hash functions employed, which is a product of the number of bands ($b$) and hashes per band ($k_{lsh}$). This yields a complexity of $O(N \cdot d \cdot b \cdot k_{lsh})$. As $b$ and $k_{lsh}$ are small, fixed hyperparameters independent of $N$, this phase runs in time effectively linear in the corpus size, i.e., $O(N \cdot d)$.

The second and dominant phase is the recursive construction of subtrees using Bisecting K-Means. The BuildSubtree function recursively partitions the data points it receives. At each level of the tree's depth, every one of the $N$ data points is processed exactly once by a K-Means clustering step (with $k = 2$). The complexity of a single K-Means operation on a partition of $n$ points is $O(n \cdot d \cdot i)$, where $i$ is the number of iterations, a small constant. Since the tree is kept approximately balanced, its depth is logarithmic with respect to the corpus size, approximately $O(\log N)$. Therefore, the total computational cost for this phase is the cost of processing all $N$ points at each of the $O(\log N)$ levels, resulting in an overall complexity of $O(N \cdot d \cdot \log N)$.

Combining the two phases, the total time complexity for building the HPT is governed by the more expensive recursive bisection step, making it $O(N \cdot d \cdot \log N)$. This is substantially more efficient than the $O(N^2 \log N)$ or greater complexity characteristic of traditional bottom-up agglomerative clustering methods, rendering our approach practical for corpora of very large scale.

### C.2  ONLINE INFERENCE OVERHEAD OF TRACE

The TRACE algorithm is engineered to introduce minimal overhead during online inference. Its efficiency stems from the fact that its computational cost is independent of the total corpus size $N$. The complexity depends on a small set of parameters: $M$, the total number of queries; $k$, the number of candidates retrieved per query; and $D_{\text{tree}}$, the maximum depth of the HPT. The total candidate pool size is thus $|S_{\text{cand}}| \leq M \cdot k$.

The online process begins by building a Prefix Tree (Trie) for each of the $M$ evidence sets, which takes $O(M \cdot k \cdot D_{\text{tree}})$ time in total, as each insertion requires traversing the depth of the tree. The core of the algorithm is the convergence depth calculation. For each of the $|S_{\text{cand}}|$ candidates, we query its path encoding against the Tries of all $M$ evidence sets. A single query to find the longest common prefix in a Trie takes $O(D_{\text{tree}})$ time. Consequently, the total complexity for this step is $O(|S_{\text{cand}}| \cdot M \cdot D_{\text{tree}})$. By substituting the upper bound for $|S_{\text{cand}}|$, this becomes $O((M \cdot k) \cdot M \cdot D_{\text{tree}}) = O(M^2 \cdot k \cdot D_{\text{tree}})$, which is the dominant component of the algorithm.

The final score calculation and sorting steps are less costly. As $M$, $k$, and $D_{\text{tree}}$ are small, bounded parameters in practice, the overall complexity of TRACE, $O(M^2 \cdot k \cdot D_{\text{tree}})$, imposes a negligible, effectively constant-time overhead at inference time. This confirms its suitability for latency-sensitive applications.

## D  ILLUSTRATIVE EXAMPLE: HIERARCHICAL ASCENT RERANKING (HAR)

To build intuition for the score-based TRACE algorithm, we present the Hierarchical Ascent Reranking (HAR) algorithm, a procedural specialization for the case of two evidence sets ($M = 2$). While

TRACE uses a declarative scoring function, HAR provides an operational view that demonstrates how prioritizing documents based on deep conceptual convergence emerges naturally from a bottom-up search. The five-stage process is illustrated step-by-step in Fig. 7.

1. **Stage 1: Initialization.** The process begins with two retrieved evidence sets, $\mathcal{S}_A = \{E, A\}$ and $\mathcal{S}_B = \{B, C\}$. The final ranked list, $R_{\text{final}}$, is initially empty. The algorithm starts its ascent from the leaf nodes of the tree that are present in these sets.

2. **Stage 2: First Ascent and Convergence.** The algorithm ascends one level. At this level, it discovers that node $F$ is the parent of both $A \in \mathcal{S}_A$ and $B \in \mathcal{S}_B$. Since $F$ is a common ancestor for documents from different sets, it is identified as a "convergence point" and is immediately added to $R_{\text{final}}$. The documents $A$ and $B$ are now accounted for by this convergence.

3. **Stage 3: Continued Ascent.** The algorithm continues ascending with the remaining active nodes. The active path from $\mathcal{S}_A$ now corresponds to node $E$ (via parent $K$), and the active path from $\mathcal{S}_B$ corresponds to node $C$ (via parent $M$). At this level, no new convergence occurs.

4. **Stage 4: Final Convergence at Root.** The algorithm reaches the root of the tree. The parents of the active nodes from the previous stage ($K$ and $M$) are both the root node, $R$. Thus, $R$ is identified as the final convergence point and is added to $R_{\text{final}}$, which now contains $\{F, R\}$. The order is critical: $F$ was found first because it is at a deeper level.

5. **Stage 5: Unpacking and Final Ranking.** The ordered list of convergence points, $R_{\text{final}} = \{F, R\}$, is unpacked. Each point is split into the original documents that caused its convergence: $F$ splits into $\{A, B\}$, and $R$ splits into $\{E, C\}$. Within each block, documents are sorted by semantic similarity to the original query. Finally, the sorted blocks are concatenated in the order their convergence points were found, yielding the final ranked list: $(A, B, E, C)$. This is then truncated to the desired top-K.

This procedural example directly illustrates the core principle of TRACE: the depth of the LCA serves as a powerful proxy for synthesis potential. Documents that converge at deeper levels (like A and B at node F) are ranked higher, as they represent a more specific and meaningful conceptual intersection.

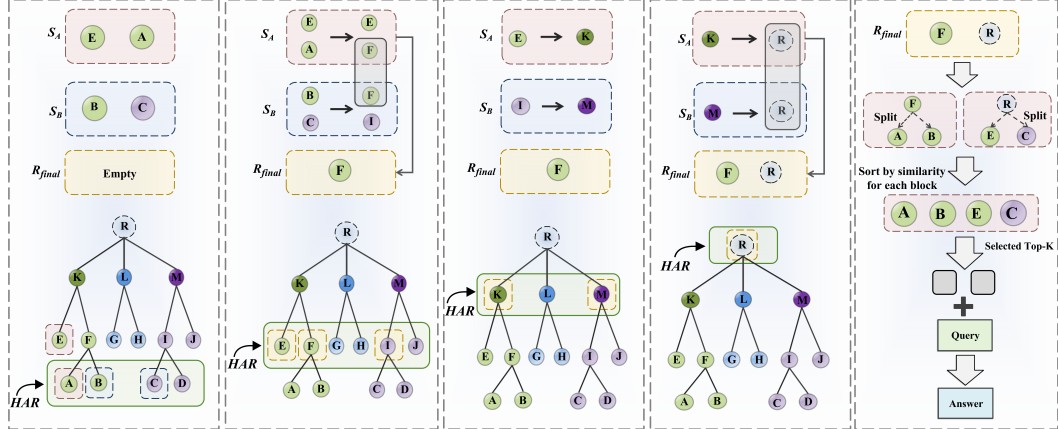

Figure 7: A Step-by-Step Walkthrough of Hierarchical Ascent Reranking (HAR). This figure illustrates the procedural execution of HAR for the special case of two evidence sets ($\mathcal{S}_A, \mathcal{S}_B$). The algorithm iteratively ascends the tree, identifying common ancestors. The rank order is determined by the level at which convergence occurs: nodes identified at deeper levels (e.g., F) are processed first and ranked higher than nodes identified at shallower levels (e.g., R).

# E    MULTI-QUERY REWRITING PROMPT

To generate a diverse set of sub-queries that probe different facets of the user's intent, our framework employs a structured multi-query rewriting strategy. This approach mandates the creation of a portfolio of four distinct query types, each designed to explore a unique semantic dimension, thereby maximizing the probability of comprehensive evidence retrieval. The design of this query portfolio is informed by recent advances in query transformation, including decompositional approaches common in multi-hop QA, "Step-Back" prompting for contextual understanding (Zheng et al., 2023), and hypothetical answer generation for improving embedding-based retrieval (Vake et al., 2025). The methodology is implemented via the following detailed prompt.

---

**System Prompt:** You are an expert query analyst for a sophisticated information retrieval system. Your task is to rewrite a given user query into EXACTLY four new queries. Each rewritten query must correspond to one of the four types defined below.

**Rewrite Types:**

1. **Decompositional Sub-Question:** Break down the original query into a smaller, self-contained, and more specific question. This sub-question should target a key factual detail required by the original query.

2. **"Step-Back" Generalization:** Generate a higher-level, more general question that provides essential context for the original query. This question should explore the broader topic or underlying principles.

3. **Perspective Reframing:** Reformulate the query from a different viewpoint or using different keywords. For instance, change a "what" question into a "how" or "why" question to uncover explanatory or procedural information.

4. **Hypothetical Scenario:** Create a concrete scenario or an example where the information sought in the original query would be highly relevant or necessary.

**Example:**

**Original Query:** "What are the limitations of method A and how does method B address them?"

**Rewritten Queries:**

1. (Decompositional) What are the specific documented weaknesses and limitations of method A?

2. (Step-Back) What are the common challenges in the problem domain that both method A and method B are designed to solve?

3. (Reframing) How does the architectural design of method B differ from method A to address its shortcomings?

4. (Hypothetical) If a system using method A consistently fails on a specific type of data, how would method B be applied to achieve a better outcome?

**User Task:**

**Original Query:** "user_query"

**Rewritten Queries:**

---

# F    EXPERIMENTAL SETUP DETAILS

## F.1    HYPERPARAMETER SETTINGS

Table 5 details the key hyperparameter values used across all experiments. These values were selected based on a combination of theoretical justification and empirical validation on development sets.

Table 5: Hyperparameter settings for HPT-TRACE experiments.

| Component | Parameter | Value |
|---|---|---|
| HPT Construction | Leaf Node Threshold ($P.leaf\_threshold$) | 30 |
| LSH | Bands ($P.lsh\_bands$) | 20 |
| | Hashes per Band ($P.lsh\_hashes\_per\_band$) | 10 |
| Retrieval & Reranking | Total Queries ($M$) | 5 |
| | Initial Candidates per Query ($P.k_{initial}$) | 15 |
| | Final Context Size ($P.k_{final}$) | 6 |

## F.2 HYPERPARAMETER JUSTIFICATION AND SENSITIVITY ANALYSIS

**Theoretical Foundation of LSH Parameter Selection.** The choice of Locality-Sensitive Hashing (LSH) parameters—bands ($b$) and hashes per band ($k$)—is fundamental to the HPT's construction. The goal is to select parameters that create a high probability of collision for similar items while minimizing it for dissimilar ones. The theoretical probability of collision for two vectors with a cosine similarity $s$ is given by $P(s) = 1 - (1 - s^k)^b$. This function produces an S-shaped curve, and an ideal curve for our purpose has a steep "transition zone" that acts as an effective similarity filter.

Figure 8 plots this function for our chosen configuration against others. Our setting ($b = 20, k = 10$) creates a steep soft filter in the desired 0.5-0.7 similarity range. This ensures that highly similar items ($s > 0.7$) almost certainly collide, while dissimilar items ($s < 0.5$) are filtered out. A less steep curve (e.g., $b = 10, k = 5$) would be too permissive, creating less coherent initial partitions. Conversely, an overly strict setting (e.g., $b = 40, k = 20$) risks permanently separating relevant items in the first step. This theoretical justification strongly supports our empirical finding that ($b = 20, k = 10$) provides an optimal balance between performance and efficiency.

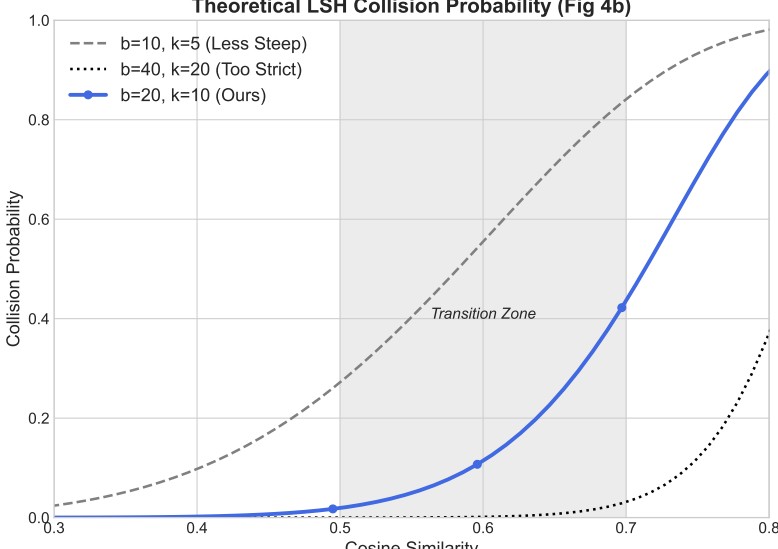

Figure 8: Theoretical LSH collision probability curves. Our chosen parameters, b=20, k=10, create a steep and well-positioned transition zone (shaded) between 0.5 and 0.7 cosine similarity, acting as an effective filter that aligns with our strong empirical results.

**Leaf Node Threshold.** The leaf_threshold parameter is a critical hyperparameter that governs the granularity of the final tree structure. It dictates the termination condition for the recursive partitioning process. A small threshold forces the algorithm to create a deep, fine-grained tree where leaf nodes represent very small and semantically tight clusters of documents. While this can enhance retrieval precision by creating highly specific evidence paths, it also significantly increases the tree's

depth and overall construction time. Furthermore, an excessively small threshold risks overfitting to the specific distribution of the training corpus, potentially creating partitions that are too specific to generalize well. Conversely, a large threshold results in a shallower, faster-to-build tree, but may group semantically distinct documents into the same leaf node. This can dilute the conceptual purity of the evidence paths and reduce the effectiveness of the TRACE algorithm, as important distinctions are lost. Our empirical analysis, presented in Table 4, confirmed this trade-off, demonstrating that a value of 30 offered the most effective balance between achieving a high F1 score and maintaining an efficient construction time.

### F.3 Implementation Details for HPT Construction

**Document Chunking.**   Prior to embedding, all corpora were segmented into smaller, manageable document chunks to serve as the leaf nodes of the HPT. We employed a sliding window approach with a fixed window size of 100 **tokens**. To preserve contextual continuity across chunk boundaries, which is crucial for capturing complete semantic units, a stride of 50 **tokens** was used. This resulted in a 50-**token** overlap between consecutive chunks, ensuring that relationships between adjacent text segments were adequately represented in the embedding space.

**Coarse-Grained Clustering via LSH.**   The construction of the HPT follows a two-phase, top-down strategy designed for scalability. The first phase employs Locality-Sensitive Hashing (LSH) to perform a **coarse-grained clustering** of the entire embedding space. Unlike centroid-based methods like K-Means, LSH functions as a form of hash-based clustering. Its objective is to group semantically similar document vectors into the same "bucket" or coarse cluster with high probability, thereby partitioning the search space efficiently.

These coarse clusters generated by LSH then serve as the initial, more manageable partitions for the second phase, where the fine-grained hierarchy is recursively built using Bisecting K-Means. We use a **Random Projection-based LSH** implementation, which is well-suited for cosine similarity. The clustering process is governed by two parameters:

- **Hashes per Band** ($k = 10$)**:** This parameter sets the strictness of the similarity condition for items to be grouped together. Within each band, two vectors must share an identical 10-bit hash signature to be considered part of the same cluster candidate.
- **Bands** ($b = 20$)**:** This parameter increases the likelihood that similar vectors will be grouped into the same coarse cluster. The algorithm considers two vectors to belong to the same partition if their hash signatures match in *at least one* of the 20 bands.

This two-phase clustering pipeline, starting with LSH for broad, efficient grouping and refining with Bisecting K-Means for local precision, is critical to the scalability of our approach. The theoretical justification for this parameter choice is detailed in Appendix F.2.

## G   Qualitative Case Studies

### G.1   Success Case: Synthesizing Disparate Factual Evidence

**Original Query:** "According to the text, why did the company's 'Project Helios' ultimately fail despite its initial success, and what role did the 'Project Chimera' leak play in its downfall?"

**Rewritten Queries:**

1. **(Decompositional)** What were the specific documented reasons for the eventual failure of Project Helios?
2. **("Step-Back" Generalization)** To understand its failure, what were the initial stated goals and successes of Project Helios?
3. **(Reframing)** How did the Project Chimera security leak causally impact the operational timeline and investor confidence in Project Helios?
4. **(Hypothetical)** If the Chimera leak had been contained, what other internal factors might have still posed a risk to Project Helios's success?

**Analysis:** This query requires synthesizing information from two distinct, though related, topics. A standard multi-query RAG system's approach reveals a critical flaw:

1. **The Semantic Reranking Pitfall:** A baseline approach would retrieve documents for all four sub-queries and merge them into a single list for reranking based on semantic similarity to the original query. This method would likely fail by prioritizing documents that are individually highly relevant but lack connective information. For instance, a detailed technical report on Helios's initial performance and an internal security memo about the Chimera incident would score highly. However, a crucial paragraph stating, "The Chimera leak exposed key Helios technology, leading to a direct loss of investor funding," might have a lower overall similarity score because it only briefly mentions each project. It would likely be outranked and missed.

2. **The Topological Solution:** HPT-TRACE fundamentally avoids this pitfall because its reranking is not driven by semantic similarity but by topological convergence. The crucial connecting paragraph, while perhaps semantically weaker in isolation, is topologically powerful. Its evidence path in the HPT would share a deep common ancestor with paths from both the "Helios" evidence sets and the "Chimera" evidence sets. The TRACE algorithm identifies this document as a point of deep convergence across multiple query facets, awarding it a high score precisely because it bridges these distinct lines of inquiry. This elevates the synthesis-critical document to the top of the ranking, demonstrating the superiority of a topology-aware approach.

### G.2 FAILURE CASE: INABILITY TO BRIDGE ABSTRACT SYMBOLIC GAPS

**Original Query:** "How does the protagonist's final acceptance of his fate symbolically connect to the 'storm' imagery from the beginning of the story?"

**Rewritten Queries:**

1. **(Decompositional)** What specific events lead to the protagonist's final acceptance of his fate?

2. **("Step-Back" Generalization)** What are the major themes and motifs used throughout the story?

3. **(Reframing)** In what ways is the 'storm' imagery used to describe characters or events in the narrative?

4. **(Hypothetical)** If the story used 'calm sea' imagery instead of a 'storm', how would the protagonist's journey be perceived differently?

**Analysi** This case highlights a key limitation by exposing a foundational assumption of our approach: that conceptual linkage is correlated with semantic proximity. The query rewriting stage functions correctly, but the topology-aware reranking mechanism fails.

1. **Topological Disconnection:** The retrieval stage would successfully gather two distinct sets of evidence. Queries 1 and 2 would retrieve passages about the protagonist's psychological journey, while Query 3 would retrieve passages containing literal descriptions of weather and scenery. However, because these two topics are semantically distant, their embeddings occupy disparate regions of the vector space. Consequently, they are placed in completely separate high-level branches of the HPT.

2. **TRACE's Blind Spot:** TRACE's scoring is predicated on structural path intersection as a proxy for conceptual connection. It has no mechanism to infer abstract, metaphorical, or symbolic connections that are not reflected in semantic similarity. Because the evidence paths for the protagonist's "psychology" and the "storm" imagery do not deeply converge, their Lowest Common Ancestor (LCA) will be very shallow (often the root node). This results in a minimal convergence depth ($c_{l,i} \approx 0$) between the two sets of documents. TRACE therefore fails to identify the relevance of the storm imagery to the character's arc, assigns it a low score, and degrades to a simple semantic similarity ranking where such documents are also deemed irrelevant.

3. **Conclusion:** This demonstrates that HPT-TRACE is optimized for synthesizing topically-related, structured knowledge where evidence can be linked through a shared conceptual hierarchy. It may struggle with queries requiring high-level abstract or symbolic reasoning across semantically dissimilar domains. Future work could explore incorporating external knowledge graphs or symbolic reasoning layers to bridge these topological gaps.

# H ADDITIONAL ROBUSTNESS TESTS

To verify that the strong performance of HPT-TRACE is a result of its core architectural design rather than an artifact of specific modeling choices, we conducted additional experiments analyzing its robustness to changes in key components.

## H.1 IMPACT OF ALTERNATIVE EMBEDDING MODELS

To verify that the strong performance of HPT-TRACE is a result of its core architectural design rather than an artifact of a specific encoder, we analyzed its robustness to changes in the embedding model. We evaluated HPT-TRACE and a Standard RAG baseline on HotpotQA using three encoders of varying quality: a powerful state-of-the-art model (BAAI/bge-large-en-v1.5), our default model (SBERT), and a smaller, faster alternative (MiniLM).

The results, presented in Table 6, reveal a critical finding. While both frameworks benefit from a stronger semantic space, the performance gap between HPT-TRACE and the baseline widens as the encoder quality improves. With the weakest encoder, HPT-TRACE provides a substantial +6.2 F1 gain; this advantage increases to +7.1 F1 with our default model and to a remarkable +7.4 F1 with the strongest encoder.

This trend demonstrates that the architectural benefit of modeling evidence path intersections is not only robust but synergistic with the quality of the underlying embeddings. As the semantic space becomes more nuanced, a simple similarity-based reranker begins to hit a performance ceiling, struggling to distinguish between many highly relevant candidates. In contrast, TRACE leverages this richer space to uncover more precise and meaningful topological connections, unlocking latent synthesis potential that the baseline cannot access. This confirms that HPT-TRACE is not merely a method that works well, but a superior architecture for evidence synthesis, especially in high-quality semantic spaces.

Table 6: Performance on HotpotQA with alternative embedding models. The architectural advantage of HPT-TRACE over Standard RAG is not only robust but widens with more powerful encoders, demonstrating its ability to unlock the latent potential of a high-quality semantic space.

| Embedding Model | HPT-TRACE (F1) | Standard RAG (F1) | $\Delta$ F1 (Gain) |
|---|---|---|---|
| **BAAI/bge-large-en-v1.5** | **79.2** | 71.8 | **+7.4** |
| SBERT / all-mpnet-base-v2 (Default) | 77.6 | 70.5 | +7.1 |
| multi-qa-MiniLM-L6-cos-v1 | 74.5 | 68.3 | +6.2 |

## H.2 IMPACT OF QUERY REWRITING STRATEGY

We conducted a rigorous evaluation of the framework's dependency on the upstream query rewriting strategy, with results on HotpotQA presented in Table 7. We compared three distinct strategies—our structured, multi-faceted approach; a simpler zero-shot prompt; and a minimal single-rewrite scenario. Crucially, these strategies were evaluated on both HPT-TRACE and a Standard RAG baseline, which merges retrieved documents and reranks them based on semantic similarity.

The results yield two key insights. First, HPT-TRACE consistently and significantly outperforms the Standard RAG baseline across all rewriting strategies, with an average performance gain of over 5 F1 points. This robust improvement demonstrates that the core performance benefit is driven by the TRACE mechanism itself. Its ability to identify synthesis-oriented documents by analyzing evidence path convergence provides a powerful signal that semantic similarity alone fails to capture.

Second, the quality and diversity of the rewritten queries amplify this architectural advantage. Within the HPT-TRACE framework, our structured rewriting approach yields a 2.5-point F1 score improvement over the zero-shot method. This is because generating queries that probe distinct semantic facets creates more varied evidence sets, providing a richer topological landscape for the TRACE algorithm to operate on and identify critical synthesis points. In conclusion, while TRACE provides a substantial performance lift under all conditions, its full potential is realized when paired with a high-quality, diverse set of initial queries.

Table 7: Ablation on query rewriting strategies on HotpotQA. HPT-TRACE consistently outperforms Standard RAG across all strategies, demonstrating that the topological reranking mechanism provides a significant and robust performance lift, which is further amplified by diverse query generation.

| Framework | Rewriting Strategy | Total Queries (M) | Answer F1 ↑ |
|---|---|---|---|
| **HPT-TRACE** | **Structured Multi-Faceted** | **5** | **77.6** |
| | Zero-shot Rewriting | 5 | 75.1 |
| | Single Rewrite | 2 | 74.2 |
| Standard RAG | Structured Multi-Faceted | 5 | 70.5 |
| | Zero-shot Rewriting | 5 | 69.7 |
| | Single Rewrite | 2 | 69.1 |

## I NOTATION

Table 8 provides a summary of the key notation used throughout the paper for easy reference.

Table 8: Notation used in the paper.

| Symbol | Description |
|---|---|
| **Sets & Structures** | |
| $\mathcal{T} = (\mathcal{V}, \mathcal{E})$ | HPT with node set $\mathcal{V}$ and edge set $\mathcal{E}$. |
| $\mathcal{D} = \{d_1, \dots, d_N\}$ | Corpus of $N$ embedded document chunks. |
| **Nodes & Embeddings** | |
| $v \in \mathcal{V}$ | A node in the tree. |
| $v_{\text{leaf}}$ | A leaf node (document) node. |
| $v_{\text{internal}}$ | An internal (structural) node. |
| $\mathcal{R}$ | The root node of the HPT. |
| $\mathbf{w}_v \in \mathbb{R}^d$ | Embedding of a leaf node $v$. |
| $p_v, \mathcal{C}_v, d_v$ | Parent, children set, and depth of node $v$. |
| **Queries & Retrieval** | |
| $Q_1$ | The original user query. |
| $\{Q_1, \dots, Q_M\}$ | The full set of $M$ queries (original + rewrites). |
| $M$ | The total number of queries. |
| $\mathcal{S}_i$ | The evidence set retrieved for query $Q_i$. |
| $S_{\text{cand}}$ | The candidate pool (union of all $\mathcal{S}_i$). |
| $k_{\text{initial}}, k_{\text{final}}$ | Initial candidate size per query and final context size. |
| **TRACE Algorithm** | |
| $\text{LCA}(l, l')$ | The Lowest Common Ancestor of nodes $l$ and $l'$. |
| $c_{l,i}$ | The convergence depth of a candidate $l$ with an evidence set $\mathcal{S}_i$. |
| $C_{\text{max}}$ | The global maximum convergence depth over the candidate pool. |
| $\text{Score}(l)$ | The final TRACE score for a candidate $l$. |

