# OpenReview forum: "Topological Retrieval-Augmented Generation via Intersecting Evidence Paths"
_ICLR.cc/2026/Conference — Submitted to ICLR 2026_

### Official Review · Reviewer_xotL · 2025-10-30

**Soundness:** 2
**Presentation:** 2
**Contribution:** 2
**Rating:** 2
**Confidence:** 4

**Summary:**

The paper presents a novel way of enhancing rag systems by restructuring the document chunks as a hierarchical tree using clustering that enables obtaining a topological score for reranking the retrieved chunks relevant to different perturbations of the query. It reduces the build time of the document strucutre tree over LLM-based RAPTOR baseline and the proposed multi-query based reranking enhances the accuracy performance over standard RAG approaches across different datasets.

**Strengths:**

- The paper is easy to follow, and the proposed algorithm is well-explained with good examples.
- Efficiency gains are obtained in the document tree building process using locally sensitive hashing and replacing LLM with a simple k-means.
- Proposed topological score using the least common ancestor is sound and novel.
- Experiments are shown on three different datasets while ablating to some components of the proposed method.

**Weaknesses:**

- Contributions are unclear as the proposed approach has many different components: (1) topological reranking, (2) document node-to-tree restructuring, and (3) Query rewriting. As (2) and (3) have been explored in prior work (just using a k-means mean representation doesn't change the performance much), (1) remains the most novel contribution. Thus, I believe the work should be positioned as a novel reranking strategy, but it is not coming out clearly in the current exposition.
- As such, TRACE does not necessitate multiple queries, but it has been introduced with additional inference-time cost and little motivation. While LLM cost to make summary nodes in RAPTOR at structuring-level was criticized, another LLM call is added to all inference steps, which is arguably even worse.
- Given the central feature being reranking, it is not compared with state-of-the-art approaches in both its efficiency and efficacy. Both semantic cross encoder and rank fusion are old works, and novel works should be included for a fair comparison especially given the fact the LLM is employed at inference time for query rewriting in the proposed method. Similarly, I do believe it should be compared against other reranking benchmarks (like TREC).
  - Ma, Yubo, et al. "Large language model is not a good few-shot information extractor, but a good reranker for hard samples!." arXiv preprint arXiv:2303.08559 (2023).
  - Dong, Jialin, et al. "Don't forget to connect! improving rag with graph-based reranking." arXiv preprint arXiv:2405.18414 (2024).
  - Tamber, Manveer Singh, Ronak Pradeep, and Jimmy Lin. "Scaling down, litting up: Efficient zero-shot listwise reranking with seq2seq encoder-decoder models." arXiv preprint arXiv:2312.16098 (2023).
  - Sun, Weiwei, et al. "Is ChatGPT good at search? investigating large language models as re-ranking agents." arXiv preprint arXiv:2304.09542 (2023).
  - Ma, Xueguang, et al. "Zero-shot listwise document reranking with a large language model." arXiv preprint arXiv:2305.02156 (2023).
- The framing of top-down and bottom-up is also problematic since both start from document nodes and do clustering to obtain the hierarchical nodes so both are bottom-up. The only difference is that RAPTOR uses an LLM to generate the summary node (and the corresponding embedding) while HPT just uses the mean representation.
- More comprehensive ablation with respect to the number of queries ($M$) should be conducted. While Table 7 is useful, it should be included in the main with other baselines to understand the trend and need for 5 different rewriting of the query.
- Query rewriting can also be done for RAPTOR and table 7 only shows for standard RAG. Since RAPTOR does not incur any additional LLM call during inference, this should be included for a fair comparison. Moreover, TRACE can be generally applied to any document tree and should thus be implemented on the RAPTOR tree as well to see plug-and-play gains.
- Results are not comprehensive as UnifiedQA is omitted from QASPER while GPT-4 is omitted from NarrativeQA. Similarly, RAPTOR is omitted from HotpotQA. These choices reduce the trust in the results. Standard RAG pipeline should also be included for all datasets and other smaller open-source LLMs should also be included. Similarly, metrics are not consistent across datasets, and it is highly recommended to include standard baselines of Rouge and F1 for all baselines.
- More advanced embedding methods in addition to mpnet should also be considered to see the sensitivity of the representational distribution.
- Minor:
  - LCA should be defined on its first occurence.
  - Figures 1, 2 and 3 should be improved for better clarity. For example, it's not clear that Q1 is same Query in Fig 3, the dark blue background makes it harder to understand in all figs, Fig 2: it's not clear how are those phases illustrated.
  - HPT-TRACE should be expanded in line 150.

**Questions:**

- What are the hyperparameters of RAPTOR?
- Why is GraphRAG not considered?
- Since baseline results are almost directly copied, how is it ensured that it is a fair comparison especially since RAPTOR's default top-k is k=5 instead of k=6 for the proposed method and the chunk sizes may differ as well? What are the examples of the correct retrieved context by HPT-TRACE as compared to RAPTOR?
- see above weaknesses

---

### Official Review · Reviewer_tz5d · 2025-11-01

**Soundness:** 1
**Presentation:** 2
**Contribution:** 1
**Rating:** 2
**Confidence:** 4

**Summary:**

The paper proposes HPT-TRACE, a framework that centers on a topology-aware reranking mechanism.

**Strengths:**

1. Originality: tree-based idea in RAG is not new anymore, but I think exploring topology sounds interesting.

2. Quality: I like Figure 1.

3. Quality: Performance scores get improved in the tables.

4. Significance: I like that the paper explores datasets of different domains such as query-focused summarization and multi-hop reasoning.

**Weaknesses:**

1. First of all, I think the soundness is low. For baselines, RAPTOR is a fairly old indexing method (almost 2-year old). It is important to demonstrate that your method is better than existing indexing methods, but it should be something better than RAPTOR. I will list 3 RAG indexing papers [1][2][3] here that claim to show better scores than RAPTOR. In fact, there are much more. I suggest the authors to use a newer method in order to provide more convincing results. Moreover, regardless of new/better baselines, I think the author should align their baselines in Table 1 and 2. Why RAPTOR is only reported in Table 1?

2. I think significance and contribution are also low. The most important thing is that performance improvement is marginal (based on a quick scan, most improvements are lower or around 1%). For Table 1, I feel that if you replace RAPTOR with something newer as I suggested, your scores might be lower. For Table 2, why RAPTOR is not used? I also see higher scores on HotpotQA in the papers I provide.

3. The paper talks a lot about RAPTOR, but I want to note that this baseline is an indexing baseline while this paper is proposing a retrieval method. This presentation confuses me and I think the overall clarity needs to be improved. It is also important to include more pure retrieval methods such as [4].

4. Regarding novelty, it is important to clearly demonstrate the benefits of exploring topology, as tree-based methods (or knowledge graph-based) are been extensively studied in the past years.

[1] "HopRAG: Multi-Hop Reasoning for Logic-Aware Retrieval-Augmented Generation"

[2] "HippoRAG: Neurobiologically Inspired Long-Term Memory for Large Language Models"

[3] "SiReRAG: Indexing Similar and Related Information for Multihop Reasoning"

[4] "RISE: Reasoning Enhancement via Iterative Self-Exploration in Multi-hop Question Answering"

**Questions:**

1. Why RAPTOR is only reported in Table 1 but not in Table 2?

2. To me, the word "falter" (line 13) seems to be confusing. Are you saying existing works are suboptimal? I guess that the sentence might come from LLMs.

---

### Official Review · Reviewer_szgC · 2025-11-02

**Soundness:** 3
**Presentation:** 3
**Contribution:** 3
**Rating:** 6
**Confidence:** 4

**Summary:**

This paper introduces HPT-TRACE, a new approach for complex question answering in RAG systems. The key insight is simple but clever. Instead of dumping all retrieved documents into one list, it builds a tree structure to organize them. Then it finds documents that bridge different parts of the question.

**Strengths:**

HPT-TRACE does something pretty different with reranking. Instead of scoring documents one by one, it looks at how they connect through a tree structure. The TRACE algorithm finds documents that bridge different parts of a question by checking how their "evidence paths" intersect.

The method might be faster because it skips expensive LLM calls when building the index. Other approaches like RAPTOR use LLMs to create summaries, which gets really slow. HPT just uses basic clustering that scales way better.

This approach actually preserves the structure from multi-query retrieval and uses it intelligently. The framework is also pretty modular - you can plug in different embedding models or query rewriting strategies.

**Weaknesses:**

The HPT building process is clever but could use more explanation about what happens under the hood. The LSH plus K-means combo sounds good in theory, but what if LSH creates really uneven buckets?

Some might be huge while others are tiny, which could mess up the tree structure. It would be helpful to know how the system handles these edge cases.

Maybe hierarchical clustering would preserve semantic relationships differently. The paper mentions a leaf threshold of 30 but doesn't really dig into how this interacts with the LSH bucket sizes or what happens if you get unlucky with the initial partitioning.

The TRACE scoring approach is interesting but raises some questions about the design choices. Why square the convergence depths in the formula? Would linear or other combinations work just as well?

The paper uses a fixed template with four specific query types (decompositional, step-back, reframing, hypothetical), but there's no real justification for why these particular categories work best.

With really large document collections, these path encodings could add up quickly. The paper also doesn't explain exactly how the tries are structured or what happens when trees get really deep and paths get long.

**Questions:**

See the weakness section for my specific technical questions.

---

### Official Review · Reviewer_9RKV · 2025-11-03

**Soundness:** 3
**Presentation:** 3
**Contribution:** 3
**Rating:** 8
**Confidence:** 3

**Summary:**

This paper addresses a critical limitation in multi-query RAG systems where retrieved documents are merged into flat lists for reranking, discarding valuable structural information. The authors propose HPT-TRACE, featuring: (1) HPT - an efficiently constructed hierarchical partition tree using top-down divisive clustering without LLMs, and (2) TRACE - a topology-aware reranking algorithm that prioritizes documents whose evidence paths deeply intersect across different query facets, measured by Lowest Common Ancestor depth. Experiments on NarrativeQA, QASPER, QuALITY, and HotpotQA show consistent improvements over SOTA methods.

**Strengths:**

1. **Important Problem Identification**: This paper points out a crucial issue that "existing methods falter by consolidating retrieved documents into a flat list for reranking," which discards valuable structural information from the multi-query rewriting process.

2. **Technical Novelty**: The paper introduces two significant innovations: HPT's efficient LLM-free construction achieving O(N·d·log N) complexity, and TRACE's topology-aware reranking based on evidence path convergence rather than isolated relevance scoring.

3. **Excellent Presentation**: Clear figures effectively illustrate complex concepts, and mathematical formulations are well-motivated. Figure 1's contrast between approaches and Figure 3's evidence path mapping are particularly helpful.

4. **Comprehensive Experimental Validation**: Extensive experiments across multiple challenging benchmarks consistently validate effectiveness, with thorough ablation studies demonstrating TRACE's critical contribution (7.1 F1 improvement).

**Weaknesses:**

1. **Strong Dependence on Semantic Similarity Assumptions**: The method assumes semantically similar documents cluster together in the HPT, which fails for queries requiring abstract, symbolic, or metaphorical reasoning where conceptual connections exist despite semantic distance.

2. **Degradation Under Topological Isolation**: When evidence paths from different sub-queries are topologically isolated, TRACE degrades to semantic similarity ranking, essentially reverting to traditional methods and undermining its core value proposition.

**Questions:**

1. **Handling Abstract Reasoning**: How might the framework address queries requiring abstract or symbolic reasoning across semantically distant domains? Have you considered incorporating external knowledge graphs to bridge topological gaps?

2. **Frequency of Degradation**: How often does topological isolation occur in practice, causing TRACE to degrade to semantic similarity? What strategies could maintain topology-aware advantages in such cases?

---

### Meta-Review · Area_Chair_embb · 2026-01-05

**Summary:**

The paper proposes HPT-TRACE, a topology-aware reranking framework for multi-query RAG that preserves structural information via a hierarchical document tree. Several reviewers find the idea intuitive and promising, with clear presentation and empirical gains on multiple QA benchmarks. However, there is significant disagreement among reviewers regarding soundness, novelty, and experimental rigor. The main concerns center on unclear positioning of the core contribution (reranking vs. indexing vs. query rewriting), reliance on outdated or misaligned baselines, inconsistent comparisons across datasets, and limited evidence that topology provides robust advantages beyond specific settings. As a result, the overall contribution is not convincingly established relative to recent RAG indexing and reranking work.

**Reviewer Concerns:**

9RKV: Positive overall; acknowledges strong motivation and technical novelty, but notes degradation when semantic similarity assumptions fail and when evidence paths are topologically isolated.

szgC: Generally positive, but raises questions about robustness of the tree construction, TRACE design choices, scalability, and insufficient ablations.

tz5d: Negative; questions soundness and significance, citing outdated baselines, marginal gains, inconsistent evaluations, and insufficient differentiation from prior tree- or graph-based RAG methods.

xotL: Negative; highlights unclear contribution boundaries, missing strong reranking baselines, inconsistent experimental settings, and limited justification for query rewriting and hierarchy construction.

**Reviewer Scores:**

9RKV: 8

szgC: 6

tz5d: 2

xotL: 2

---

### Decision · Program_Chairs · 2026-01-26

Reject